# Silver-Catalyzed Decarboxylative Acylation of Isocyanides Accesses to α-Ketoamides with Air as a Sole Oxidant

**DOI:** 10.3390/molecules28145342

**Published:** 2023-07-11

**Authors:** Jia Xu, Xue Li, Xing-Yu Chen, Yu-Ting He, Jie Lei, Zhong-Zhu Chen, Zhi-Gang Xu

**Affiliations:** College of Pharmacy, National & Local Joint Engineering Research Center of Targeted and Innovative Therapeutics, IATTI, Chongqing University of Arts and Sciences, Chongqing 402160, China; xujia_0907@163.com (J.X.); lixue91jlsp@sina.com (X.L.); faith0909@163.com (X.-Y.C.); jjjxu_92@126.com (Y.-T.H.); jlei0916@163.com (J.L.)

**Keywords:** silver catalyst, decarboxylation, air oxidation, isocyanide, α-ketoamide

## Abstract

α-Ketoamide moieties, as privileged units, may represent a valuable option to develop compounds with favorable biological activities, such as low toxicity, promising PK and drug-like properties. An efficient silver-catalyzed decarboxylative acylation of α-oxocarboxylic acids with isocyanides was developed to derivatize the α-ketoamide functional group via a multicomponent reaction (MCR) cascade sequence in one pot. A series of α-ketoamides was synthesized with three components of isocyanides, aromatic α-oxocarboxylic acid analogues and water in moderate yields. Based on the research, the silver-catalyzed decarboxylative acylation confirmed that an oxygen atom of the amide moiety was derived from the water and air as a sole oxidant for the whole process.

## 1. Introduction

α-Ketoamides are privileged units and widely exist in biologically relevant molecules, natural products, pharmaceuticals and other functional materials [1,2]. The α-ketoamide functional group is also a key part of market drugs and drug candidates with favorable biological activities, promising PK and low toxicity, thus helping to face biological targets of increased complexity [3,4,5]. The favorable biological activities of α-ketoamides have been deeply exploited by modifying their structural rigidity or by conferring their capacity to increase hydrogen bonds, leading to the improvement of their pharmaceutical profile. Through continuing modification, their peculiar properties make α-ketoamides a privileged structure in medicinal chemistry that has led to the development of a wide array of compounds that have shown a variety of pharmacological activities [6,7,8]. In recent years, medicinal chemists have constantly exploited α-ketoamides to modify molecules with clinical potential, primarily as sedative/hypnotics, anxiolytics, antitumorals, antibacterials, antivirals and antiprions. For instance, their derivatives are essential constituents in inhibitors, agonist and bioactive compounds (Figure 1) [1,2,3,4,5,6,7,8]. Furthermore, the α-ketoamides’ moiety also serves as a versatile and valuable intermediate and synthon that provides the gateway to access a variety of useful scaffolds in a number of functional group transformations and total syntheses. Owing to their importance and broad applications, numerous synthetic methods for the preparation of α-ketoamides have been developed over the past few decades [9,10,11,12,13,14,15,16].

Recently, transition metal-catalyzed decarboxylative transformations of α-oxocarboxylic acids have gained considerable attention due to their advantages of broad functional group compatibility, readily available feedstocks, simple operations and moderate to good yields [17,18,19,20,21,22,23,24]. Through the great efforts chemists have made, strategies have been established, and a direct approach was found to afford the acyl group. Prompted by these results, a decarboxylative acyl group insertion into the useful synthons to provide *N*-monosubstituted α-ketoamides was developed. For example, Wang’s group developed direct a copper-catalyzed decarboxylative acylation to generate α-ketoamides [25]. This cross-coupling reaction was performed between the acyl C-H of formamides and α-oxocarboxylic acids using 2 equiv. of DTBP as the oxidant and 2 equiv. of PivOH as an additive (Figure 1A). Patel and coworkers reported palladium-catalyzed chemoselective insertion into organic cyanamides via decarboxylation. Meanwhile, 2 equiv. of (NH_4_)_2_S_2_O_8_ is necessary to directly access α-ketoamides (Figure 1B) [26]. Although these impressive contributions had been made, more efficient and practical catalytic systems for the synthesis of α-ketoamide are still highly attractive and in demand.

In the last 30 years, chemists have become more and more preoccupied about “green” sequences of reactions to produce a myriad of high-value chemicals [27]. So, atmospheric molecular oxygen is an ideal sequence because it is safe and free [28]. Neither transition metal-catalyzed decarboxylation for the synthesis of α-ketoamides has used oxygen as an oxidant. The development of alternative sustainable routes from commercially available starting materials and oxygen to access α-ketoamides is highly warranted. Isocyanides represent one of the most important and versatile synthons due to their unique electronic and structural characteristics. Moreover, isocyanide-based multicomponent reaction rapidly facilitated complex molecules in one pot from simple starting materials with remarkable synthetic efficiency and high-atom economy [29,30,31,32,33]. Based on the continuous interests in silver-catalyzed organic reactions of isocyanide and the Hunsdiecker reaction [34,35,36], we attempted to develop an alternative transition metal-catalyzed decarboxylation to construct α-ketoamides with isocyanide using air as the only oxidant. Herein, we report an efficient and practical silver-catalyzed decarboxylation to rapidly facilitate α-ketoamides using air as the sole oxidant (Figure 1C). The importance of the given chemistry is threefold: (a) operation simplicity; (b) avoided additional oxidant (without stoichiometric peroxide or persulfide; (c) suppressed competitive reaction to achieve precise synthesis (no oxazole by-products were observed).

## 2. Results

To begin our investigation for silver-catalyzed decarboxylation to synthesize α-ketoamides, phenylglyoxylic acid and isocyanocyclohexane were selected as the model substrates in a solvent of acetonitrile (MeCN). In the screening of silver salts (Table 1, entries 1–6), we rapidly found that AgOTf was a powerful catalyst to afford α-ketoamides with a yield of 70%. To increase the conversion of final product **4a**, solvents were evaluated, such as *N,N*-dimethylformamide (DMF), 1,4-dioxane, 1,2-dichloroethane (DCE), dimethyl sulfoxide (DMSO), tetrahydrofuran (THF) and toluene (entries 7–12). Among the screened solvents, DCE led to a higher yield of 82%. When the reaction was performed at 100 °C for 1 h, **4a** was isolated with a yield of 87% (entry 13). Fortunately, prolonging the reaction time to 2 h delivered α-ketoamide **4a** with the highest yield of 89% (entry 15). To the best of our knowledge, a stoichiometric oxidant was necessary to direct the silver-catalyzed decarboxylation of α-oxocarboxylic acids. Switching our attention to oxidants of PhI(OAc)_2,_ K_2_S_2_O_8_ and oxone, diminished yields of **4a** were found. Therefore, oxygen was considered as the ideal oxidant in our protocol. To confirm the efficiency of the silver catalyst, the other transition metals were also investigated. None of them afforded the desired product **4a** with a satisfactory yield (entries 19–21). When optimizing the amount of water, we found that a 2.0 equiv. of water was more suitable to our system. Thus, the optimized reaction conditions of the three-component, one-pot reaction were determined to be: “**1a** (0.3 mmol), **2a** (0.3 mmol), **3** (2.0 equiv.) with 10 mol% AgOTf as catalyst by using DCE under 100 °C for 2 h.”

After establishing the optimal reaction conditions, we set out to explore the substrate generality of the silver-catalyzed decarboxylative acylation of isocyanides to access α-ketoamides. Various phenylglyoxylic acids and isocyanides were subjected to this one-pot decarboxylative process, as the results show in Figure 2, and the high tolerance of substituents on the substrates was observed. Under standard reaction conditions, the desired product **4a** was isolated with a yield of 89% and unequivocally confirmed by X-ray crystallography (CCDC 2267589). Specifically, the reaction worked well not only with aliphatic isocyanides but also with aromatic isocyanides. Despite the steric effect of isocyanides, the transformations for synthesizing α-ketoamides using 1,1,3,3-tetramethylbutyl isocyanide or *tert*-butyl isocyanide as an amide source did not decrease the reaction yields (**4c**, 89%; **4d**, 89%; **4e**, 88%). As the literature reported [37], the reaction with α-oxocarboxylic acids and toluenesulfonylmethyl isocyanide (TosMIC) was notably prone to giving the corresponding 5-phenyloxazole. Under optimal reaction conditions, the starting materials were directly converted to compound **4f** with a 90% yield. The aromatic isocyanides of 2,6-dimethylphenyl isocyanide, 2-chloro-6-methylphenyl isocyanide and 2-isocyanonaphthalene, 4-methoxyphenyl isocyanide, 4-bromophenyl isocyanide and 4-chlorophenyl isocyanide afforded desired compounds in a good yield ranging from 79% to 86%. To expeditiously expand chemical diversity and to reach novel chemical space, α-oxo-1,3-benzodioxole-5-acetic acid was employed using as an acyl source. Various isocyanides were subjected to the standard conditions; comparatively, α-oxocarboxylic acid analogues containing electron-donating group generated lower yields of final products (**4o**–**4x**). Similar results were found for the substrates of 2-thiopheneglyoxylic acid and 2-furanglyoxylic acid, as the corresponding starting reagents were converted into the desired α-ketoamides with moderate yields (**4y**–**4ac**, 49–61%). Unfortunately, alkyl α-oxocarboxylic acid could not be tolerated in sliver-catalyzed decarboxylation with air as the sole oxidant, as depicted in Figure 2 (**4ad**–**4af**).

As relative studies have reported, decarboxylation with α-keto acids as an acyl source always undergoes a radical process [25,26]. To further verify the possibility, 2.0 equiv. of TEMPO (common radical scavenger) was added into the mixture of phenylglyoxylic acid, TosMIC and water. The formation of α-ketoamide **4f** was dramatically inhibited. Unsurprisingly, TEMPO-adduct **5f** was observed by ESI-HRMS under standard reaction conditions (Figure 3A). To clarify the mechanism by which isocyanide was used as an amide source, the model reaction was performed in the absence of water, and a trace amount of **4f** was obtained (Figure 3B). The reaction was performed under a nitrogen atmosphere (Figure 3C). As the results show, oxygen was essential to start the decarboxylation of α-keto acids. The results indicate that AgOTf was oxidized by oxygen to generate Ag(II) species.

Based on the control experiments and previous reports, the possible reaction mechanism was elucidated in Figure 4. First, the decarboxylation of α-keto acids was initiated by AgOTf that was oxidized by air. Subsequently, the acyl radical was formed by releasing 1 mole of CO_2_. Then, isocyanide trapped the in situ-generated radical species **7** to furnish the important intermediate **8**. Finally, the final compound was generated by the nucleophilic attack of H_2_O on the nitrilium intermediate **9**. It could be found that acyl radical was a key intermediate to afford the final product. When α-ketoacids were directly conjugated with aromatic groups, aromatic radicals were formed which were more stable than aliphatic radicals. That is why aliphatic α-oxocarboxylic acids could not proceed well under standard reaction conditions.

## 3. Materials and Methods

### 3.1. Analytical Techniques

^1^H and ^13^C NMR were recorded on a Bruker 400 spectrometer. ^1^H NMR data are reported as follows: chemical shift in ppm (δ), multiplicity (s = singlet, d = doublet, t = triplet, m = multiplet), coupling constant (Hz), relative intensity. ^13^C NMR data are reported as follows: chemical shift in ppm (δ). LC/MS analyses were performed on a Shimadzu-2020 LC-MS instrument using the following conditions: Shim-pack VP-ODS C18 column (reverse phase, 150 × 4.6 mm); a linear gradient from 10% water and 90% acetonitrile to 75% acetonitrile and 25% water over 6.0 min; flow rate of 0.5 mL/min; UV photodiode array detection from 200 to 400 nm. High-resolution mass spectra (HRMS) were recorded on a Q Exactive hybrid quadrupole-Orbitrap mass spectrometer (Thermo Scientific) with an ESI source of 140,000 fwhm, AGC target set to 1 × 10^6^, and a scan range of 100–1000 *m*/*z*. The raw data were deconvoluted using an Xcalibur 4.1. UV-VIS spectrophotometer TU-1950. The products were purified by Biotage Isolera™ Spektra Systems and hexane/EtOAc solvent systems. All reagents and solvents were obtained from commercial sources and used without further purification.

### 3.2. Synthetic Procedures for the Synthesis of Compound ***4***

α-Oxocarboxylic acids (0.3 mmol), isocyanide (0.3 mmol), water (2.0 equiv.), and AgOTf (10 mol%) were mixed in DCE (2.0 mL). The reaction mixture was performed in a 5.0 mL microwave vial and stirred at 100 °C for 2 h. After completion, the reaction mixture was monitored by TLC, and the solvent was removed and extracted with ethyl acetate. Then, the organic phase was dried through Mg_2_SO_4_ and concentrated. The residue was purified by silica gel column chromatography using a gradient of ethyl acetate/hexane (0–100%) to afford the relative targeted product **4**.

*N*-Cyclohexyl-2-oxo-2-phenylacetamide (**4a**). White solid (89%), R_f_ = 0.30 (n-hexane/EtOAc, 8:2). ^1^H NMR (400 MHz, CDCl_3_) δ 8.35 (d, J = 8.3 Hz, 2H), 7.66–7.61 (m, 1H), 7.49 (t, J = 7.6 Hz, 2H), 6.99 (s, 1H), 3.96–3.80 (m, 1H), 2.06–1.96 (m, 2H), 1.83–1.62 (m, 4H), 1.47–1.37 (m, 2H), 1.31–1.25 (m, 2H). ^13^C NMR (100 MHz, CDCl_3_) δ 188.15, 160.86, 134.32, 133.45, 131.22, 128.46, 48.49, 32.73, 25.43, 24.76. HRMS (ESI) calcd for C_14_H_18_NO_2_^+^ ([M + H]^+^) 232.1332, Found 232.1336.

*N*-Cyclopentyl-2-oxo-2-phenylacetamide (**4b**). White solid (90%), R_f_ = 0.30 (n-hexane/EtOAc, 8:2). ^1^H NMR (400 MHz, CDCl_3_) δ 8.27 (d, J = 7.4 Hz, 2H), 7.54 (t, J = 7.4 Hz, 1H), 7.40 (t, J = 7.8 Hz, 2H), 6.97 (s, 1H), 4.27–4.17 (m, 1H), 2.03–1.96 (m, 2H), 1.70–1.63 (m, 2H), 1.61–1.56 (m, 2H), 1.49–1.41 (m, 2H). ^13^C NMR (100 MHz, CDCl_3_) δ 187.99, 161.31, 134.32, 133.44, 131.24, 128.46, 51.22, 32.96, 23.81. HRMS (ESI) calcd for C_13_H_16_NO_2_^+^ ([M + H]^+^) 218.1176, Found 218.1177.

*N*-(Adamantan-1-yl)-2-oxo-2-phenylacetamide (**4c**). White solid (89%), R_f_ = 0.33 (n-hexane/EtOAc, 8:2). ^1^H NMR (400 MHz, CDCl_3_) δ 8.29–8.25 (m, 2H), 7.59–7.53 (m, 1H), 7.46–7.39 (m, 2H), 6.84 (s, 1H), 2.09 (s, 9H), 1.70 (s, 6H). ^13^C NMR (100 MHz, CDCl_3_) δ 188.61, 160.94, 134.11, 133.41, 131.20, 131.10, 128.34, 128.26, 52.35, 41.07, 40.98, 36.23, 36.15, 29.35, 29.27. HRMS (ESI) calcd for C_18_H_22_NO_2_^+^ ([M + H]^+^) 284.1645, Found 284.1645.

*N*-(*tert*-Butyl)-2-oxo-2-phenylacetamide (**4d**). White solid (89%), R_f_ = 0.33 (n-hexane/EtOAc, 8:2). ^1^H NMR (400 MHz, CDCl_3_) δ 8.27–8.20 (m, 2H), 7.56–7.49 (m, 1H), 7.39 (t, J = 7.7 Hz, 2H), 6.86 (s, 1H), 1.39 (s, 9H). ^13^C NMR (100 MHz, CDCl_3_) δ 188.60, 161.13, 134.16, 133.41, 131.22, 128.39, 51.68, 28.40. HRMS (ESI) calcd for C_12_H_16_NO_2_^+^ ([M + H]^+^) 206.1176, Found 206.1177.

2-Oxo-2-phenyl-*N*-(2,4,4-trimethylpentan-2-yl)acetamide (**4e**). White solid (88%), R_f_ = 0.33 (n-hexane/EtOAc, 8:2). ^1^H NMR (400 MHz, CDCl_3_) δ 8.28–8.20 (m, 2H), 7.53 (dd, J = 10.6, 4.3 Hz, 1H), 7.39 (t, J = 7.8 Hz, 2H), 6.94 (s, 1H), 1.76 (s, 2H), 1.44 (s, 6H), 0.97 (s, 9H). ^13^C NMR (100 MHz, CDCl_3_) δ 188.58, 160.73, 134.12, 133.43, 131.26, 128.37, 55.52, 51.71, 31.72, 31.45, 28.74. HRMS (ESI) calcd for C_12_H_16_NO_2_^+^ ([M + H]^+^) 262.1802, Found 262.1806.

2-Oxo-2-phenyl-*N*-(tosylmethyl)acetamide (**4f**). White solid (90%), R_f_ = 0.20 (n-hexane/EtOAc, 8:2). ^1^H NMR (400 MHz, CDCl_3_) δ 8.07 (d, J = 8.1 Hz, 2H), 7.83 (d, J = 8.3 Hz, 2H), 7.66–7.62 (m, 1H), 7.45 (t, J = 7.8 Hz, 2H), 7.35 (d, J = 8.1 Hz, 2H), 4.80 (d, J = 7.0 Hz, 2H), 2.45 (s, 3H). ^13^C NMR (100 MHz, CDCl_3_) δ 186.03, 160.82, 145.71, 134.93, 133.43, 132.51, 131.09, 130.11, 129.67, 129.08, 128.59, 126.53, 60.00, 21.74. HRMS (ESI) calcd for C_12_H_16_NO_2_^+^ ([M + H]^+^) 318.0795, Found 318.0796.

*N*-(4-Fluorobenzyl)-2-oxo-2-phenylacetamide (**4g**). A white solid (90%), R_f_ = 0.25 (n-hexane/EtOAc, 8:2). ^1^H NMR (400 MHz, CDCl_3_) δ 8.49–8.30 (m, 2H), 7.70–7.60 (m, 1H), 7.50 (t, J = 7.8 Hz, 3H), 7.39–7.30 (m, 2H), 7.13–6.99 (m, 2H), 4.55 (d, J = 6.1 Hz, 2H). ^13^C NMR (100 MHz, CDCl_3_) δ 187.51, 163.59, 161.59, 161.14, 134.58, 133.24, 133.00, 132.97, 131.26, 129.71, 129.63, 128.57, 115.85, 115.63, 42.77. HRMS (ESI) calcd for C_15_H_13_FNO_2_^+^ ([M + H]^+^) 258.0925. Found:258.0928.

*N*-(2,6-Dimethylphenyl)-2-oxo-2-phenylacetamide (**4h**). White solid (83%), R_f_ = 0.25 (n-hexane/EtOAc, 8:2). ^1^H NMR (400 MHz, CDCl_3_) δ 8.50–8.38 (m, 3H), 7.69 (t, J = 7.4 Hz, 1H), 7.54 (t, J = 7.8 Hz, 2H), 7.21–7.14 (m, 3H), 2.33 (s, 6H). ^13^C NMR (100 MHz, CDCl_3_) δ 187.72, 159.86, 135.15, 134.69, 133.23, 132.46, 131.40, 128.65, 128.40, 127.81, 18.52. HRMS (ESI) calcd for C_16_H_16_NO_2_^+^ ([M + H]^+^) 254.1176, Found 254.1180.

*N*-(2-Chloro-6-methylphenyl)-2-oxo-2-phenylacetamide (**4i**). White solid (85%), R_f_ = 0.25 (n-hexane/EtOAc, 8:2). ^1^H NMR (400 MHz, CDCl_3_) δ 8.67 (s, 1H), 8.46–8.40 (m, 2H), 7.68 (t, J = 7.4 Hz, 1H), 7.54 (t, J = 7.8 Hz, 2H), 7.35 (dd, J = 6.7, 2.6 Hz, 1H), 7.25–7.17 (m, 2H), 2.37 (s, 3H). ^13^C NMR (100 MHz, CDCl_3_) δ 187.06, 159.66, 137.77, 134.71, 133.12, 131.41, 129.39, 128.64, 128.42, 127.32, 19.05. HRMS (ESI) calcd for C_15_H_13_ClNO_2_^+^ ([M + H]^+^) 274.0629, Found 274.0631.

*N*-(Naphthalen-2-yl)-2-oxo-2-phenylacetamide (**4j**). White solid (81%), R_f_ = 0.20 (n-hexane/EtOAc, 8:2). ^1^H NMR (400 MHz, CDCl_3_) δ 9.06 (s, 1H), 8.41–8.35 (m, 3H), 7.79 (dd, J = 8.4, 3.6 Hz, 2H), 7.75 (d, J = 8.0 Hz, 1H), 7.60 (t, J = 7.4 Hz, 1H), 7.53 (dd, J = 8.8, 2.1 Hz, 1H), 7.46 (t, J = 6.0 Hz, 2H), 7.44–7.41 (m, 1H), 7.40–7.36 (m, 1H). ^13^C NMR (100 MHz, CDCl_3_) δ 187.32, 159.00, 134.68, 134.05, 133.77, 133.15, 131.54, 131.13, 129.13, 128.60, 127.89, 127.67, 126.80, 125.59, 119.53, 117.14. HRMS (ESI) calcd for C_18_H_14_NO_2_^+^ ([M + H]^+^) 276.1019, Found 276.1020.

*N*-(4-Methoxyphenyl)-2-oxo-2-phenylacetamide (**4k**). White solid (86%), R_f_ = 0.25 (n-hexane/EtOAc, 8:2). ^1^H NMR (400 MHz, CDCl_3_) δ 8.89 (s, 1H), 8.48–8.41 (m, 2H), 7.70–7.62 (m, 3H), 7.53 (t, J = 7.8 Hz, 2H), 6.98–6.93 (m, 2H), 3.85 (s, 3H). ^13^C NMR (100 MHz, CDCl_3_) δ 187.61, 158.68, 157.12, 134.54, 133.26, 131.46, 129.83, 128.54, 121.53, 114.41, 55.51. HRMS (ESI) calcd for C15H14NO3+ ([M + H]+) 256.0968, Found 256.0971.

*N*-(4-Bromophenyl)-2-oxo-2-phenylacetamide (**4l**). White solid (83%), Rf = 0.20 (n-hexane/EtOAc, 8:2). 1H NMR (400 MHz, CDCl3) δ 8.92 (s, 1H), 8.38–8.29 (m, 2H), 7.61–7.57 (m, 1H), 7.56–7.51 (m, 2H), 7.46–7.41 (m, 4H). 13C NMR (100 MHz, CDCl3) δ 187.05, 158.80, 135.73, 134.83, 132.91, 132.27, 131.51, 128.65, 121.46, 118.10. HRMS (ESI) calcd for C14H11BrNO2+ ([M + H]+) 303.9968, Found 303.9968.

*N*-(4-Chlorophenyl)-2-oxo-2-phenylacetamide (**4m**). White solid (79%), Rf = 0.20 (n-hexane/EtOAc, 8:2). 1H NMR (400 MHz, CDCl3) δ 8.92 (s, 1H), 8.33 (d, J = 8.1 Hz, 2H), 7.62–7.56 (m, 3H), 7.44 (t, J = 7.8 Hz, 2H), 7.29 (d, J = 8.8 Hz, 2H). 13C NMR (100 MHz, CDCl3) δ 187.08, 158.79, 135.22, 134.83, 132.93, 131.51, 130.42, 129.33, 128.65, 121.15. HRMS (ESI) calcd for C14H11ClNO2+ ([M + H]+) 260.0473, Found 260.0475.

*N*-(Furan-2-ylmethyl)-2-oxo-2-phenylacetamide (**4n**). White solid (90%), Rf = 0.30 (n-hexane/EtOAc, 8:2). 1H NMR (400 MHz, CDCl3) δ 8.38 (dd, J = 8.2, 0.9 Hz, 2H), 7.66 (t, J = 7.4 Hz, 1H), 7.58–7.38 (m, 4H), 6.50–6.20 (m, 2H), 4.61 (d, J = 5.9 Hz, 2H). 13C NMR (100 MHz, CDCl3) δ 187.34, 161.42, 150.14, 142.63, 134.52, 133.27, 131.27, 128.55, 110.57, 108.12, 36.39. HRMS (ESI) calcd for C13H12NO3+ ([M + H]+) 230.0812, Found 230.0813.

2-(Benzo[*d*][1,3]dioxol-5-yl)-*N*-(4-chlorophenyl)-2-oxoacetamide (**4o**). White solid (64%), Rf = 0.30 (n-hexane/EtOAc, 7:3). 1H NMR (400 MHz, CDCl3) δ 8.93 (s, 1H), 8.22 (dd, J = 8.3, 1.6 Hz, 1H), 7.80 (d, J = 1.6 Hz, 1H), 7.60–7.55 (m, 2H), 7.30–7.27 (m, 2H), 6.85 (d, J = 8.3 Hz, 1H), 6.02 (s, 2H). 13C NMR (100 MHz, CDCl3) δ 159.16, 153.58, 148.16, 135.30, 130.33, 129.58, 129.30, 127.34, 121.12, 110.61, 108.28, 102.14. HRMS (ESI) calcd for C15H11ClNO4+ ([M + H]+) 304.0371, Found 304.0375.

2-(Benzo[*d*][1,3]dioxol-5-yl)-*N*-(4-methoxyphenyl)-2-oxoacetamide (**4p**). White solid (59%), Rf = 0.30 (n-hexane/EtOAc, 7:3). 1H NMR (400 MHz, CDCl3) δ 8.82 (s, 1H), 8.22 (dd, J = 8.3, 1.7 Hz, 1H), 7.80 (d, J = 1.6 Hz, 1H), 7.56–7.51 (m, 2H), 6.88–6.82 (m, 3H), 6.01 (s, 2H), 3.75 (s, 3H). 13C NMR (100 MHz, CDCl3) δ 185.06, 159.05, 157.03, 153.34, 148.06, 129.88, 129.44, 127.77, 121.51, 114.37, 110.63, 108.21, 102.07, 55.52. HRMS (ESI) calcd for C16H14NO5+ ([M + H]+) 300.0866, Found 300.0870.

2-(Benzo[*d*][1,3]dioxol-5-yl)-*N*-(4-fluorophenyl)-2-oxoacetamide (**4q**). White solid (50%), Rf = 0.30 (n-hexane/EtOAc, 7:3). 1H NMR (400 MHz, CDCl3) δ 8.91 (s, 1H), 8.21 (dd, J = 8.3, 1.7 Hz, 1H), 7.79 (d, J = 1.6 Hz, 1H), 7.58 (dd, J = 9.1, 4.7 Hz, 2H), 7.01 (t, J = 8.7 Hz, 2H), 6.83 (d, J = 8.3 Hz, 1H), 6.01 (s, 2H). ^13^C NMR (100 MHz, CDCl_3_) δ 184.70, 159.17, 153.49, 148.12, 132.81, 129.51, 127.58, 121.70, 121.62, 116.07, 115.85, 110.60, 108.25, 102.12. HRMS (ESI) calcd for C_15_H_11_FNO_4_^+^ ([M + H]^+^) 288.0667, Found 288.0670.

2-(Benzo[*d*][1,3]dioxol-5-yl)-*N*-(2,6-dimethylphenyl)-2-oxoacetamide (**4r**). White solid (65%), Rf = 0.30 (n-hexane/EtOAc, 7:3). 1H NMR (400 MHz, CDCl3) δ 9.02 (s, 1H), 7.49 (dd, J = 8.2, 1.7 Hz, 1H), 7.35 (s, 1H), 7.17 (s, 1H), 7.09 (d, J = 7.2 Hz, 2H), 6.86 (d, J = 8.1 Hz, 1H), 6.04 (s, 2H), 2.16 (s, 6H). 13C NMR (100 MHz, CDCl3) δ 168.03, 161.91, 154.00, 148.88, 136.39, 131.16, 129.90, 128.86, 127.76, 108.58, 108.36, 102.47, 18.01. HRMS (ESI) calcd for C17H16NO4+ ([M + H]+) 298.1074, Found 298.1077.

2-(Benzo[*d*][1,3]dioxol-5-yl)-*N*-(tert-butyl)-2-oxoacetamide (**4s**). White solid (53%), Rf = 0.30 (n-hexane/EtOAc, 8:2). 1H NMR (400 MHz, CDCl3) δ 8.17–8.09 (m, 1H), 7.74 (t, J = 3.3 Hz, 1H), 6.97 (s, 1H), 6.85 (dd, J = 8.3, 3.9 Hz, 1H), 6.04 (s, 2H), 1.44 (s, 9H). 13C NMR (100 MHz, CDCl3) δ 186.21, 161.54, 152.92, 147.89, 128.97, 127.89, 110.37, 108.01, 101.95, 51.60, 28.36. HRMS (ESI) calcd for C13H16NO4+ ([M + H]+) 250.1074. Found 250.1076.

2-(Benzo[*d*][1,3]dioxol-5-yl)-*N*-(4-methoxybenzyl)-2-oxoacetamide (**4t**). White solid (56%), Rf = 0.35 (n-hexane/EtOAc, 7:3). 1H NMR (400 MHz, CDCl3) δ 8.22 (dd, J = 8.3, 1.7 Hz, 1H), 7.81 (d, J = 1.6 Hz, 1H), 7.40 (s, 1H), 7.32–7.21 (m, 2H), 6.94–6.84 (m, 3H), 6.08 (s, 2H), 4.50 (d, J = 6.0 Hz, 2H), 3.82 (s, 3H). 13C NMR (100 MHz, CDCl3) δ 185.24, 161.85, 159.27, 153.18, 148.02, 129.29, 129.13, 127.95, 114.23, 110.35, 108.14, 102.01, 55.32, 42.97. HRMS (ESI) calcd for C17H16NO5+ ([M + H]+) 314.1023, Found 314.1024.

2-(Benzo[*d*][1,3]dioxol-5-yl)-*N*-(furan-2-ylmethyl)-2-oxoacetamide (**4u**). White solid (61%), Rf = 0.30 (n-hexane/EtOAc, 8:2). 1H NMR (400 MHz, CDCl3) δ 8.24 (dd, J = 8.3, 1.7 Hz, 1H), 7.83 (d, J = 1.6 Hz, 1H), 7.47 (s, 1H), 7.42 (dd, J = 1.8, 0.7 Hz, 1H), 6.91 (d, J = 8.3 Hz, 1H), 6.38–6.33 (m, 2H), 6.10 (s, 2H), 4.59 (d, J = 5.9 Hz, 2H). 13C NMR (100 MHz, CDCl3) δ 184.89, 161.74, 153.27, 150.17, 148.05, 142.61, 129.21, 127.84, 110.55, 110.35, 108.18, 108.07, 102.05, 36.38. HRMS (ESI) calcd for C14H12NO4+ ([M + H]+) 274.0710, Found 274.0713.

*N*-(Adamantan-1-yl)-2-(benzo[*d*][1,3]dioxol-5-yl)-2-oxoacetamide (**4v**). White solid (65%), Rf = 0.30 (n-hexane/EtOAc, 8:2). 1H NMR (400 MHz, CDCl3) δ 8.16 (dd, J = 8.3, 1.7 Hz, 1H), 7.78 (d, J = 1.6 Hz, 1H), 6.86 (t, J = 10.2 Hz, 2H), 6.07 (s, 2H), 2.12 (t, J = 8.3 Hz, 9H), 1.73 (s, 6H). 13C NMR (100 MHz, CDCl3) δ 186.25, 161.18, 152.91, 147.90, 129.02, 127.96, 110.47, 108.02, 101.95, 52.32, 41.10, 36.25, 29.35. HRMS (ESI) calcd for C19H22NO4+ ([M + H]+) 328.1543, Found 328.1544.

2-(Benzo[*d*][1,3]dioxol-5-yl)-*N*-(naphthalen-2-yl)-2-oxoacetamide (**4w**). White solid (48%), Rf = 0.30 (n-hexane/EtOAc, 7:3). 1H NMR (400 MHz, CDCl3) δ 9.19 (s, 1H), 8.44 (d, J = 1.7 Hz, 1H), 8.36 (dd, J = 8.3, 1.7 Hz, 1H), 7.93 (d, J = 1.5 Hz, 1H), 7.86 (dd, J = 15.7, 8.3 Hz, 3H), 7.61 (dd, J = 8.8, 2.1 Hz, 1H), 7.55–7.44 (m, 2H), 6.95 (d, J = 8.3 Hz, 1H), 6.11 (s, 2H). ^13^C NMR (100 MHz, CDCl_3_) δ 184.76, 159.40, 153.46, 148.13, 134.12, 133.76, 131.08, 129.55, 129.09, 127.88, 127.67, 126.77, 125.54, 119.56, 117.06, 110.67, 108.26, 102.12. HRMS (ESI) calcd for C_19_H_14_NO_4_^+^ ([M + H]^+^) 320.0917, Found 320.0917.

*N*-(4-Chlorophenyl)-2-(4-methoxyphenyl)-2-oxoacetamide (**4x**). White solid (49%), R_f_ = 0.30 (n-hexane/EtOAc, 8:2). ^1^H NMR (400 MHz, CDCl_3_) δ 8.98 (s, 1H), 8.46–8.41 (m, 2H), 7.61–7.57 (m, 2H), 7.31–7.27 (m, 2H), 6.93–6.89 (m, 2H), 3.84 (s, 3H). ^13^C NMR (100 MHz, CDCl_3_) δ 184.82, 165.07, 159.40, 135.37, 134.34, 130.25, 129.28, 125.97, 121.11, 114.02, 55.65. HRMS (ESI) calcd for C_15_H_13_ClNO_3_^+^ ([M + H]^+^) 290.0578. Found 290.0580.

2-(Furan-2-yl)-2-oxo-*N*-(2,4,4-trimethylpentan-2-yl)acetamide (**4y**). White solid (54%), R_f_ = 0.30 (n-hexane/EtOAc, 8:2). ^1^H NMR (400 MHz, CDCl_3_) δ 8.10 (d, J = 3.6 Hz, 1H), 7.67 (d, J = 1.0 Hz, 1H), 7.16 (s, 1H), 6.54 (dd, J = 3.6, 1.6 Hz, 1H), 1.73 (s, 2H), 1.42 (s, 6H), 0.95 (s, 9H). ^13^C NMR (100 MHz, CDCl_3_) δ 174.70, 159.05, 149.46, 149.18, 126.76, 113.08, 55.39, 51.70, 31.71, 31.42, 28.68. HRMS (ESI) calcd for C_14_H_22_NO_3_^+^ ([M + H]^+^) 252.1594, Found 252.1596.

*N*-Cyclopentyl-2-(furan-2-yl)-2-oxoacetamide (**4z**). White solid (61%), R_f_ = 0.30 (n-hexane/EtOAc, 8:2). ^1^H NMR (400 MHz, CDCl_3_) δ 8.22 (d, J = 3.6 Hz, 1H), 7.78 (d, J = 0.9 Hz, 1H), 7.27 (s, 1H), 6.65 (dd, J = 3.6, 1.5 Hz, 1H), 4.36–4.21 (m, 1H), 2.11–2.03 (m, 2H), 1.80–1.66 (m, 4H), 1.59–1.50 (m, 2H). ^13^C NMR (100 MHz, CDCl_3_) δ 173.87, 159.66, 149.52, 149.30, 126.87, 113.13, 51.17, 32.87, 23.81. HRMS (ESI) calcd for C_11_H_14_NO_3_^+^ ([M + H]^+^) 208.0968, Found 208.0970.

2-(Furan-2-yl)-*N*-(4-methoxybenzyl)-2-oxoacetamide (**4aa**). White solid (58%), R_f_ = 0.30 (n-hexane/EtOAc, 8:2). ^1^H NMR (400 MHz, CDCl_3_) δ 8.23 (d, J = 3.6 Hz, 1H), 7.78 (d, J = 0.8 Hz, 1H), 7.62 (s, 1H), 7.27 (d, J = 8.6 Hz, 2H), 6.90 (d, J = 8.6 Hz, 2H), 6.65 (dd, J = 3.5, 1.5 Hz, 1H), 4.50 (d, J = 6.0 Hz, 2H), 3.82 (s, 3H). ^13^C NMR (100 MHz, CDCl_3_) δ 173.61, 159.91, 159.33, 149.52, 149.42, 129.30, 129.08, 126.92, 114.26, 113.17, 55.32, 42.94. HRMS (ESI) calcd for C_14_H_14_NO_4_^+^ ([M + H]^+^) 260.0917, Found 260.0918.

*N*-(4-Fluorophenyl)-2-(furan-2-yl)-2-oxoacetamide (**4ab**). White solid (49%), R_f_ = 0.30 (n-hexane/EtOAc, 8:2). ^1^H NMR (400 MHz, CDCl_3_) δ 9.13 (s, 1H), 8.31 (d, J = 3.7 Hz, 1H), 7.92–7.80 (m, 1H), 7.78–7.64 (m, 2H), 7.15–7.11 (m, 2H), 6.71 (dd, J = 3.7, 1.6 Hz, 1H). ^13^C NMR (100 MHz, CDCl_3_) δ 173.38, 161.21, 158.77, 157.53, 149.84, 127.51, 121.78, 121.70, 116.14, 115.91, 113.41. HRMS (ESI) calcd for C_12_H_9_FNO_3_^+^ ([M + H]^+^) 234.0561, Found 234.0564.

*N*-Benzyl-2-oxo-2-(thiophen-2-yl)acetamide (**4ac**). White solid (52%), R_f_ = 0.30 (n-hexane/EtOAc, 8:2). ^1^H NMR (400 MHz, CDCl_3_) δ 9.08 (s, 1H), 7.80 (d, J = 4.9 Hz, 1H), 7.66 (d, J = 3.7 Hz, 1H), 7.32 (d, J = 7.1 Hz, 2H), 7.28–7.21 (m, 3H), 7.11 (t, J = 4.4 Hz, 1H), 4.94 (s, 2H). ^13^C NMR (100 MHz, CDCl_3_) δ 179.67, 167.29, 162.34, 139.21, 138.28, 137.31, 135.52, 129.07, 128.71, 128.69, 128.06, 43.27. HRMS (ESI) calcd for C_13_H_12_NO_2_S^+^ ([M + H]^+^) 246.0583, Found 246.0585.

The results of the X-ray diffraction analysis for compound **4a** were deposited with the Cambridge Crystallographic Data Centre (CCDC 2267589) (Appendix A).

## 4. Conclusions

In summary, we uncovered a one-pot MCR of silver-catalyzed decarboxylative acylation with α-oxocarboxylic acids, isocyanides and water. A series of α-ketoamides was synthesized, with air as the sole oxidant participating in the decarboxylative process. The control experiments confirmed that the oxygen atom of the amide moiety was derived from water. Notably, transition metal-catalyzed decarboxylation using oxygen as oxidants for the synthesis of α-ketoamides has not been well explored to date. And isocyanides, as one of the most important and versatile synthons, could achieve decarboxylative acylation but not directly acylated with carboxylic acid. The available data suggest that the introduction of cheap, handily accessible α-oxocarboxylic acids into decarboxylative acylation reactions might exhibit a powerful alternative to synthesize drug-like α-ketoamides.

## Data Availability

Not applicable.

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
