# Peer review of "Silver-Catalyzed Decarboxylative Acylation of Isocyanides Accesses to α-Ketoamides with Air as a Sole Oxidant"

_molecules, 2023, doi:10.3390/molecules28145342_

Round 1

Reviewer 1 Report

The manuscript “Decarboxylative acylation of isocyanides accesses to α-ketoamides” reports a new procedure to pharmaceutically important precursors. In substance, the novelty of the work is in changing an oxidant to air, which is promising from the view of green chemistry. The chemistry reported is quiet standard. I think the manuscript suits to Molecules after major revision.

1. The title should be more specified.

2. The authors should describe why they used 2equiv. of H2O. Why the amount of water was not optimized.

3. The experiment with 18O was performed incorrectly. 18O isotope can be present in the product due to simple exchange processes not connected to the reaction. The authors can check it just holding the 16O product in the presence of 18O water and checking the crude reaction mixture by MS. More informative are 17O experiments with NMR, but they are expensive and difficult. I recommend just to delete experiments with 18O.

4. Line 200: The reaction mixture was sealed… What amount of air was there? Does the amount of air in the vessel influence the yield?

5. “suppressed competitive reaction to achieve precise synthesis” what does it mean? A discussion on it is required.

α-Ketoamides are privileged units widely exists in biologically relevant molecules, 

the reaction was performed with 100 86 oC for 1 h, sightly increased yield of 4a was obtained

yields was obtained. Therefore, oxygen was think of the idea oxidant in our protocol. To

gated. None of them afford desired product 4a with satisfied yield (entries 19-21). Thus, 

confirm the efficiency of silver catalyst, the others transition metals were also investi

conditions, the starting materials were directly converted to targeted compound 4f with

equiv. of TEMPO (common radical scavengers) was added into the mixture of phenyl

to access α-ketoamide under nitrogen atmosphere were explored (Scheme 3D). As the

Author Response

Dear reviewers,

Please check the attached file for all the answers.

best

Zhigang

Reviewer 2 Report

Reviewer comments:

  The present work (molecules-2490065), reported by Jia et.al., entitled “Decarboxylative acylation of isocyanides accesses to 2 α-ketoamides” deals with the novel synthetic methodology of biologically important targets and their characterization in an attractive manner. This work is having broader substrate scope, uses eco & economically friendly oxidant i.e., water and operationally simple experimental procedure was optimized. Based on these points, the present manuscript can be accepted with the following minor revisions.

1.       Authors have mentioned the use of DCE. If it is so, then how this process will be considered as a “greener process”? Authors have to justify this in their revised manuscript.

2.       In page 4, line 136-137, both aliphatic and aromatic nitriles were resulted good yields of the product. Authors have mentioned only aromatic α-oxocarboxylic acids but this methodology is not extendible to aliphatic α-oxocarboxylic acids. Authors have to justify this further in their revised manuscript.

3.       In the title, “access” word is more suitable instead of “accesses”.

4.       Authors have to mention the influence of relative stoichiometric ratios of three reactants in this process.

5.       The importance of multicomponent reactions (MCRs) and One-Pot synthesis were not described in the main body of the manuscript and it was mentioned only in abstract and in conclusion. Authors have to describe these concepts and the following recent citations have to be cited.

i)                    https://doi.org/10.1016/j.rechem.2021.100202

ii)                   https://doi.org/10.2174/1570178620666230111103902

iii)                 https://doi.org/10.1002/slct.202202636

iv)                 https://doi.org/10.1016/j.tetasy.2011.08.011

Author Response

(The authors gave the same response as above.)

Reviewer 3 Report

This paper described about the efficient synthesis of α-ketoamides by the reaction of α-keto acids with isocyanides involving decarbonylation of α-keto acids.

The experiments to elucidate the reaction mechanism are appropriate, and the reaction mechanism proposed by the authors is judged to be appropriate and also synthetically useful.

After the following remarks are corrected, I consider it acceptable for publication.

(1) Since this reaction seems to be a decarbonylation similar to the Hunsdiecker Reaction, the paper on the Hunsdiecker Reaction should be cited.

(2) Page 3, Third line of Results: In the screening of silver salts, the term "common organic acid" is used, but silver salts are not organic acids.

(3) Although the authors emphasize that it is a green reaction, it seems possible to synthesize the corresponding α-ketoamide from the α-keto acid and amine using appropriate condensing agents. I am suspicious as to which reaction is greener.

(4) Scheme 2: It needs to be commented why compound 4ad~4af could not be synthesized.

Author Response

(The authors gave the same response as above.)

Round 2

Reviewer 1 Report

The authors have provided all required revision to the manuscript. In my opinion, it is ready for the acceptance.